# Optical Diffractive Convolutional Neural Networks Implemented in an All-Optical Way

**DOI:** 10.3390/s23125749

**Published:** 2023-06-20

**Authors:** Yaze Yu, Yang Cao, Gong Wang, Yajun Pang, Liying Lang

**Affiliations:** 1School of Artificial Intelligence, Hebei University of Technology, Tianjin 300401, China; 202032803060@stu.hebut.edu.cn; 2Center for Advanced Laser Technology, Hebei University of Technology, Tianjin 300401, China; yang.cao@hebut.edu.cn (Y.C.); wanggong@hebut.edu.cn (G.W.); 3Hebei Key Laboratory of Advanced Laser Technology and Equipment, Tianjin 300401, China

**Keywords:** diffraction effect, 4*f* system, convolutional neural network, image classification

## Abstract

Optical neural networks can effectively address hardware constraints and parallel computing efficiency issues inherent in electronic neural networks. However, the inability to implement convolutional neural networks at the all-optical level remains a hurdle. In this work, we propose an optical diffractive convolutional neural network (ODCNN) that is capable of performing image processing tasks in computer vision at the speed of light. We explore the application of the 4*f* system and the diffractive deep neural network (D${}^{2}$NN) in neural networks. ODCNN is then simulated by combining the 4*f* system as an optical convolutional layer and the diffractive networks. We also examine the potential impact of nonlinear optical materials on this network. Numerical simulation results show that the addition of convolutional layers and nonlinear functions improves the classification accuracy of the network. We believe that the proposed ODCNN model can be the basic architecture for building optical convolutional networks.

## 1. Introduction

Image recognition is an important application of artificial neural networks (ANNs). Expressive feature information can be obtained by extracting the intrinsic connections of image data [1,2]. After years of development, it has been applied to many fields, such as medical diagnosis, intelligent systems, and assisted driving [3,4,5]. However, the massive amount of data brought by neural networks has caused a huge computational burden and energy consumption to electronic computing [6,7]. Compared with electrons, photons feature faster speed, more substantial information-carrying capacity, higher parallelism, as well as anti-interference [8,9,10]. Therefore, the optical neural network is expected to be used to solve the problem of insufficient computing power caused by the failure of Moore’s Law [11,12,13,14].

In 2018, Lin et al. proposed an all-optical diffraction deep neural network (D${}^{2}$NN) [15], which loads image information into phase, amplitude, and other features with optical diffraction effects to achieve the connection between the input and output layers. This revealed the capability of all-optical D${}^{2}$NN in classifying MNIST and Fashion–MNIST datasets. Sun et al. fitted the saturated absorption coefficient of nonlinear optical materials and introduced a nonlinear activation function on the basis of D${}^{2}$NN [16]. Dou et al. proposed a Res-D${}^{2}$NN, which enables the training of deeper diffractive networks by constructing diffractive residual learning blocks [17]. During the experiment, we found that the classification effectiveness of D${}^{2}$NN was not satisfactory in dealing with the CIFAR-10 dataset that was constructed using natural scenes. This inadequacy highlights the model’s limitations in feature extraction. In electronic neural networks, D${}^{2}$NN can be regarded as a classifier composed of multiple fully connected layers whose core is the product of matrix vectors or linear combinations of neurons. The neurons in the fully connected layers cannot determine the critical classification features containing tremendous redundant information in a transfer process. Therefore, convolution is proposed as a feature extractor in deep learning [18,19]. The convolutional layer can complete the effective feature extraction in advance of classification, thus filtering redundant information. The convolution kernel, with a much smaller size than the image, maps the high-dimensional raw data to the low-dimensional hidden layer feature space. Then, the fully connected layer completes the mapping of distributed features to the sample labeling space.

For realizing optical convolution, the 4*f* system is a better choice. It is well-known that a lens can perform the two-dimensional Fourier transform at the speed of light [20]. Therefore, an optical 4*f* system can be used instead of conventional processors to achieve complicated convolutional operations, thereby reducing computational costs. Chang et al. constructed an efficient hybrid optoelectronic neural network by combining the 4*f* system and electronic fully connected layer and verified the feasibility of the 4*f* system as a convolutional layer [21]. Furthermore, Colburn et al. improved the classification performance of the AlexNet network using the 4*f* system as an optical front-end to preprocess the input data [22]. In these works, the convolutional operation was implemented optically, and the results of optical calculations were fed into the electronic network. The optical–electronic conversion module undoubtedly suffers additional computing costs in its electronic part, thus resulting in failure in accelerating the neural network.

Therefore, this study proposes an optical diffractive convolutional neural network (ODCNN). We combine the 4*f* system with D${}^{2}$NN to optically implement convolutional neural networks. The classification accuracy of this network is verified through simulations on the MNIST and Fashion–MNIST datasets. Then, we analyze its capability in classifying the CIFAR-10 dataset. Finally, the effect of nonlinear units on the network is explored. This network forgoes the participation of electrons and fully exploits optics computational efficiency and parallel processing capability. The proposed ODCNN demonstrates insight into the computational process in optical neural networks and opens up opportunities to implement all-optical convolutional neural networks. More importantly, it fully considers the feasibility of hardware implementation. Thus, the proposed framework can find practical applications in machine vision and promote the cross-fertilization of optics and computing disciplines.

## 2. Model and Principles of ODCNN

Figure 1 shows the sketch of the ODCNN, which is a neural network framework based on an all-optical implementation. The two primary modules correspond to the convolutional layer and the fully connected layer found in an electronic convolutional neural network. The input image passes through an optical convolution layer consisting of an array of lenses, i.e., the 4*f* optical correlators. The front focal plane of the first lens and the back focal plane of the second lens are the input and output planes of the convolutional layer, respectively. The corresponding filter masks are set in the Fourier plane. In this case, the optical transformation or processing of the input optical field signal can be completed rapidly with the assistance of the 4*f* optical system [23]. The feature map is obtained and then input to the fully connected layer for feature weighting. D${}^{2}$NN completes the mapping from the feature space to the sample marker space. D${}^{2}$NN is composed of superimposed optical diffraction layers. Each grid on the diffraction layer corresponds to a pixel and a neuron in the neural network. The neuron can adjust the phase of the incident light and propagate backward as a secondary wave source, and the connection between neurons can be represented by the Rayleigh–Sommerfeld diffraction equation [24]. The classification results are finally detected in the output layer of D${}^{2}$NN in light intensity distribution, depending on the image type of the classification. In the following sections, we delve into the detailed implementation of each part.

### 2.1. Convolutional Layer

The convolution operation in neural networks typically follows the input image, and each convolutional layer consists of a set of learnable convolution kernels. These kernels can be viewed as visual filters with fixed sizes and certain depths, determined by the dimensions of the input data. During the convolution operation, the convolution kernels slide over the image with a set stride to filter out non-correlated regions, highlight the convergent spaces with correlation, and compress the information space while retaining effective information. In optical processing systems, the required convolution by neural networks can be executed in the Fourier domain using a simpler dot-product method, and the 4*f* system can achieve the transformation from real space to Fourier space.

The system consists of two convex lenses with equal focal lengths, as shown in Figure 2. The first lens, L1, is located at a distance of *f* from the incident plane. The incident beam of light converges after passing through the lens and forms an image at the back focal plane, known as the Fourier plane. Another lens, L2, is placed at a distance of *f* from this plane. In the back focal plane of L2, an inverted conjugate image is formed at a total distance of 4*f* from the original target plane. The input image undergoes Fourier transformation after passing through the lens and is then convolved with a kernel. Afterwards, it undergoes inverse Fourier transformation, resulting in the convolution output.

The accuracy of image classification is related to the multi-dimensional features of the image. Complex image classification tasks require plenty of image features. The number of convolutional kernels indicates the number of feature maps output by that convolutional layer. Therefore, it is necessary to set up multiple convolution kernels for image operations. Following image convolution operations, feature maps from conventional convolutional networks and optical 4*f* systems exhibit notable distinctions in their output. The core of the 4*f* system is the filter on the frequency spectrum, which affects the image output of the entire system. Usually, a single filter is employed, corresponding to a sole convolutional kernel, thereby producing just a single feature map. The 4*f* lens array cannot physically realize the deep stacked arrangement of convolutional kernels. Therefore, it is necessary to design a filter to achieve the output of multiple feature maps.

In this study, a phase filter was employed to only modify the relative phase distribution of each frequency component without affecting the energy of the incident light field, thus exhibiting high optical efficiency. By tiling and arranging N × N convolution kernels on the spectrum plane, the convolution results of the input image and multiple two-dimensional kernels can be obtained. This results in a horizontally tiled output image. The point spread function was used to describe the light field distribution of the system in the form of an impulse response. The convolutional plane can be described as follows:(1)$$PSF\left(x,y\right)=\sum_{m=1}^{N}\sum_{n=1}^{N}Kernel_{(m-1)N+n}\left(x,y\right)*\delta \left(x-\Delta x,y-\Delta y\right).$$

The function $Kernel_{(m-1)N+n}\left(x,y\right)$ represents the weight distribution of the ${\left(\left(m-1\right)N+n\right)}^{th}$ convolution kernel. Optimizing the layout of convolution kernels by adjusting Δx and Δy can prevent crosstalk between output feature maps in the system model. The parameter design of the plane mainly comprises the size and quantity of the convolutional kernels. Increasing the number of convolutional kernels can lead to an improvement in the network’s maximum performance; however, an excessive number can also cause performance fluctuations and impede the convergence speed. On the other hand, the size of the convolutional kernels can significantly influence the efficacy of feature extraction.

Following extensive simulation studies and continuous parameter adjustments, we determined that the optimal number of convolutional kernels is 16, with a size of 9 × 9 pixels. Fine-tuning the spacing between the convolutional kernels based on the specific dataset is feasible to ensure the effectiveness of the resulting feature maps. This can be achieved using either a spatial light modulator or a fixed phase mask. During the simulation, we initialized the convolution kernel by tiling and zero-padding it. Then, we processed the input image signal in the frequency domain. Figure 3 shows a selection of some flattened convolutional kernels, thereby elucidating the distribution and output of two distinct convolutional methods.

The spreading processing is necessary for feature maps distributed in a stacked manner when entering the fully connected layer. The multichannel 2-D matrix is transformed into a 1-D vector by vertical separation, horizontal splicing, and multiplication by the corresponding weight coefficients at each neuron operation. Similarly, neurons modulate their amplitude or phase when the tiled feature map passes through the diffractive layer in the form of incident light. The spatial structural properties of the feature map are ignored in this process. Therefore, it is not necessary to consider the output form of the feature map, which can directly process through the fully connected layer, whether stacked or tiled.

### 2.2. Fully Connected Layer

Figure 4 shows the structure of the fully connected layer. The fully connected layer is composed of a diffractive deep neural network. As a completely passive optical computing system, each plane consisting of diffraction gratings constitutes a layer of the neural network. The multiple layers of the network are interconnected through coherent light. The network modulates the input light waves through multiple diffraction planes to perform all-optical inference and image classification tasks. Experimental evidence has demonstrated that training can be accomplished using deep learning methods such as stochastic gradient descent and error backpropagation.

The feature maps are fed into the network in the form of phase or amplitude encoding. Each grid in the diffractive layer can be viewed as a neuron, which modulates the input light waves through diffraction effects. The neuron serves as a secondary wave source to transmit the propagation process to the lower layers of the network. The connection weight between neurons in the network can be expressed by the Rayleigh–Sommerfeld diffraction formula:(2)$$w_{i}^{l}\left(x,y,z\right)=\frac{z-z_{i}}{r^{2}}\left(\frac{1}{2\pi r}+\frac{1}{j\lambda }\right)exp\left(\frac{j2\pi r}{\lambda }\right).$$
where *l* denotes the *l*th layer of the network, *i* represents the *i*th neuron located at position $\left(x_{i},y_{i},z_{i}\right)$ in the *l*th layer of the network, and *r* denotes the Euclidean distance between the current neuron and the corresponding neuron in the adjacent diffractive layer. The network employs two modulation methods, namely, amplitude and phase modulation, and, thus, the modulation coefficient of the neuron can be obtained as follows:(3)$$t_{i}^{l}\left(x_{i},y_{i},z_{i}\right)=\alpha_{i}^{l}\left(x_{i},y_{i},z_{i}\right)exp\left[j\phi_{i}^{l}\left(x_{i},y_{i},z_{i}\right)\right],$$
where $\alpha_{i}^{l}\left(x_{i},y_{i},z_{i}\right)$ represents the amplitude modulation factor and $\phi_{i}^{l}\left(x_{i},y_{i},z_{i}\right)$ represents the phase modulation factor. These two factors can be used separately or combined for complex modulation [25].

The diffractive propagation process in the network is shown in Figure 5. For a fully linear optical network, the entire process from data input, through network propagation, to final output can be regarded as a matrix operation of a network layer. However, it is difficult for a single diffractive layer to perform the mathematical operations completed by multiple diffractive planes, so it instead provides a basis for the depth advantage of multi-layer networks. To compare the differences between diffractive networks and fully connected neural networks (FCNN), the neuron propagation function of FCNN is given:(4)$$y_{i}^{l}=\Sigma_{k}\omega_{k,i}^{l-1}x_{k}^{l-1}+b_{i}^{l},$$
where $\omega_{k,i}^{l-1}$ and $b_{i}^{l}$ are learnable parameters. The connections between neurons are independent of each other. From Equation (Equation 2), it can be observed that the weight depends only on the distance between the input wavelength and the neuron. Therefore, the weight $w_{i}^{l}$ can be regarded as a constant before the network starts training. The true weight parameters are the amplitude and phase factors in Equation (Equation 3). All inputs from the upper layer of the network correspond to the same amplitude and phase parameters on the neuron. This indicates weight sharing among the connections in the diffractive network. In addition, the neuron propagation function in FCNN is a linear function, which results in weak mapping ability for complex feature spaces. The multi-layer mapping transformation process between input and output can be compressed into a single operation. Therefore, it is generally necessary to add a nonlinear activation function to adapt to complex nonlinear mapping. In the diffractive deep neural network, the optical path propagation between multiple diffractive planes is difficult to achieve through a single diffraction, and it possesses certain nonlinear characteristics without the introduction of a nonlinear function.

In the output layer, a pre-defined category area is partitioned on the output plane. The intensity information within the area is used to represent the probability distribution of the classification results. For example, the CIFAR-10 dataset containing 10 classes of images corresponds to 10 pre-defined areas on the output plane. Then, the detectors are used to measure the intensity signals. Different categories of input images have different spatial relationship features. After multiple layers of diffraction effects, they present irregular bright and dark images. Through training, the output intensity results of the images can correspond to the pre-defined areas. The goal is to maximize the signal intensity within the corresponding areas while minimizing the influence of signals outside the areas on the classification results. In the experiment, the softmax function was used to map the output of the fully connected layer to a range of 0 to 1. The maximum output of the softmax layer corresponded to the maximum intensity signal, which is the predicted image category of the network.

## 3. Simulation Results and Analysis

We conducted experiments on three datasets for image classification, namely, MNIST, Fashion–MNIST, and CIFAR-10, and compared the classification accuracy with the D${}^{2}$NN benchmark network to evaluate the performance of the ODCNN structure. In addition, considering the possibility of optical implementation of the network, batch normalization, pooling, and dropout operations were not incorporated into the experiments.

In the network, a convolutional layer containing a standard 16-channel 9 × 9 convolutional kernel and a 5-layer diffraction fully connected layer containing 5 × 200 × 200 neurons were designed. According to the characteristics of the 4*f* system, the 16-channel convolutional kernels were tiled in space. At the same time, the 5-layer diffractive fully connected layer can exhibit good classification performance of D${}^{2}$NN without convolutional operations. Table 1 shows the distribution of pixels in the model. Taking the MNIST dataset as an example, the input image of 28 × 28 pixels size was expanded to 200 × 200 with zero padding. In the convolution layer, 16 convolution kernels were discretized into a 4 × 4 array and tiled into a 200 × 200 size planar space, as shown in Figure 6. Then, the input image was convolved with the tiled plane. The output feature map directly entered the fully connected layer, consisting of five diffraction layers for feature reduction. Finally, we obtained the result of the intensity distribution of the output plane.

The stochastic gradient descent method was used in training, and the parameters in the network were optimally adjusted using the Adam optimizer [26]. The loss function used was the mean square error:(5)$$MSE=\frac{1}{n}\sum_{i=1}^{n}{\left(Y_{i}-\hat{Y_{i}}\right)}^{2},$$
where $Y_{i}$ is the true probability distribution and $\hat{Y_{i}}$ is the predicted value. The obtained error results were backpropagated, and the phase modulation coefficients in the network were iteratively updated in the process of minimizing the loss results. The batch size set for the training process was 64. Each dataset was subjected to a total of 10,000 iterations on a server (NVIDIA 2080Ti Graphical Processing Unit) for about 5 h, using Python version 3.8.10 and TensorFlow framework version 2.3.0.

The MNIST and Fashion–MNIST datasets provide a unified evaluation standard for the validation of new models. Their simple data format facilitates the presentation of model feasibility and validity effects. We evaluated and analyzed the performance of the networks on the MNIST dataset, with the results shown in Figure 7. The loss curves of the two networks are almost identical, maintaining the same convergence rate and, eventually, converging to zero. Both networks achieved good classification results on this dataset. The accuracy curve shows that the final classification accuracy of the ODCNN reached 97.32%, which is significantly improved compared with the value of 92.35% of the basic D${}^{2}$NN network.

This improvement in accuracy is also reflected for the Fashion–MNIST dataset, as shown in Figure 8. After adding the convolutional layer, ODCNN improved the classification accuracy from 82.1% to 86.75%. As well, the convergence of the loss curve was faster. After 2500 iterations, the ODCNN reached convergence, while the benchmark network required 4000 iterations. Compared with the 5-layer diffraction network, the classification accuracy of ODCNN was improved to different degrees on both datasets, which proves the necessity of feature extraction in image classification tasks. The convergence speed of the loss curve also indicates that ODCNN can determine the best combination of parameters faster.

Further, we chose the more complex CIFAR-10 dataset to verify the impact of the convolutional layer on the classification performance, which consists of 3-channel RGB images of real-world objects with more noise and complex scale features. Since the system was programmed to handle monochromatic light, the images were grayscaled before training, turning them into one-dimensional images. Figure 9 shows the classification results of the two networks on the ten classes of images in this dataset using confusion matrices. The relative distribution of the confusion matrices of the two networks was the same, i.e., there was a limitation in predicting a certain class of images. However, the number of misclassifications, such as erroneously labeling “2” and “4” images as “0” and misclassifying “3” images as “5”, significantly decreased compared to previous instances. Although ODCNN fails to change the relative distribution of results, the ability of the convolutional structure to extract useful information reduces its overall classification error rate. On this dataset, the overall accuracy increased from 39.4% to 49.35%.

To reflect the classification effect of the two networks on the ten types of images in the CIFAR-10 dataset, Figure 10 and Figure 11 show the precision and recall rate corresponding to each type of image, respectively. The precision refers to the number of correctly classified images in each type of result, reflecting the accuracy of the network. The recall reflects the ability of the model to find positive samples. The ODCNN achieved a higher precision for all ten classes of images, with an average improvement of about 10%. In terms of recall rate, ODCNN also achieved an overall advantage. However, when we focused on one of the image categories, the two indicators did not improve by the same amount. Depending on the classification task, precision and recall have different emphases, but, in this experiment, the two indicators were equally important. Therefore, we chose a more robust indicator, the ROC curve, to compare the performance of the two models.

When the sample distribution is imbalanced, the ROC curve can more stably reflect the quality of the model. The curve takes the False Positive Rate (FPR) and True Positive Rate (TPR) as the horizontal and vertical coordinates. It consists of the performance of multiple classifiers on test samples. Therefore, the ROC curve characterizes how the classifier performance varies with the threshold value. In addition, the size of the area under it is called AUC, which can quantitatively describe the difference between classifiers. As shown in Figure 12, the ROC curves of ODCNN were all above D${}^{2}$NN. The ODCNN tends to achieve higher TPR with the same threshold, which means it provides more positive predictions. In addition, the AUC of ODCNN and D${}^{2}$NN were 0.53 and 0.46, respectively, which indicates that ODCNN has a higher accuracy. The AUC of D${}^{2}$NN on CIFAR-10 was less than 0.5, and its prediction accuracy was even lower than random prediction. The addition of the convolutional layer improved the AUC of the model, making it predictive when a reasonable threshold was set.

The optical convolutional layer improves the classification performance by enhancing the feature extraction capability of the network. This change is especially evident in complex data. Nonetheless, the fully linear operation of the network weakens representation and limits further performance improvement. The linearity is insufficient for specific functions of a neural network. In contrast, the nonlinear function can improve the convergence speed of the network and recognition accuracy, which is an indispensable part of the neural network [27]. Otherwise, regardless of the number of network layers, it can be summarized into a vast linear operation. Meanwhile, most problems are nonlinear. Therefore, it is of necessity to add a nonlinear activation function to ODCNN. Enhancing the nonlinear expression of the network can facilitate the convolutional layer to obtain more complex image features. We introduced a linear rectification unit(ReLU) [28] after the 4*f* system: (6)$$f\left(x\right)=max\left[0,g\left(x\right)\right],$$
where g(x) is the linear correction *x*. The linear correction also circumvents the effect of weak noise on the results and reduces the overall computational cost of the network.

However, the simulation took into account the constraints of the system, namely the non-existence of negative values of light intensity in physics. In this case, the traditional ReLU function cannot effectively perform truncation. Therefore, we proposed a new non-linear function expression in our experiment:(7)$$f\left(x\right)=\left\{\begin{array}{c} 0\;x<\alpha \cdot 255\\ x\;x\geq \alpha \cdot 255 \end{array}\right.,$$
where *x* represents the light intensity and α represents the threshold coefficient. When the input light intensity surpasses the threshold intensity, the transmission proceeds normally; conversely, the output light intensity decreases to zero. The optimal size of the threshold coefficient varies depending on the dataset. We characterized the improved ODCNN using the Fashion–MNIST dataset, and the threshold coefficient was 0.4. The results in Figure 13 illustrate that the nonlinear ODCNN achieved higher accuracy and improved convergence speed. The nonlinear layer can be realized by adjusting the camera curve of sCMOS [29]. Moreover, composite materials such as photorefractive crystals(SBN:60) [30] show excellent nonlinear properties in the optical field, promising the implementation of nonlinear functions in optical neural networks.

After the aforementioned experiments, the feasibility and effectiveness of ODCNN have been thoroughly validated. To demonstrate the model’s capability in image classification, Table 2 compares the classification accuracy of various state-of-the-art optical network models with ODCNN on the MNIST and Fashion–MNIST datasets. The proposed network in this paper has a more concise structure but achieves comparable accuracy to the current top-performing optical networks. It achieves relatively better network performance at a smaller computational cost. The table also includes the experimental results of an electronic neural network (ENN) with a consistent structure to ODCNN, indicating that optical networks still lag slightly behind electronic networks in terms of performance. However, considering the differences in energy consumption and cost, this gap can be considered negligible.

## 4. Conclusions

In this paper, we developed a forward propagation model ODCNN based on the optical 4*f* system and diffractive fully connected layer. We explored the similarities and differences between this model and traditional convolutional neural networks. The network was trained using three datasets, namely, MNIST, Fashion–MNIST, and CIFAR-10. The hyperparameters of the model were determined by random search, and the network was optimized by the Adam optimizer. The classification accuracy on the three datasets—97.32%, 86.75%, and 49.35%—reveals that the ODCNN with the convolution layer outperformed the benchmark network. In addition, the introduction of the nonlinear unit enhanced the accuracy of this network. We believe that the appearance of the optical convolutional layer can improve the classification performance of the network. The combination of optical convolutional layer and optical diffractive neural network can open the path to the realization of the optical convolutional neural network in the imminent future. 

## Figures and Tables

**Figure 1 sensors-23-05749-f001:**
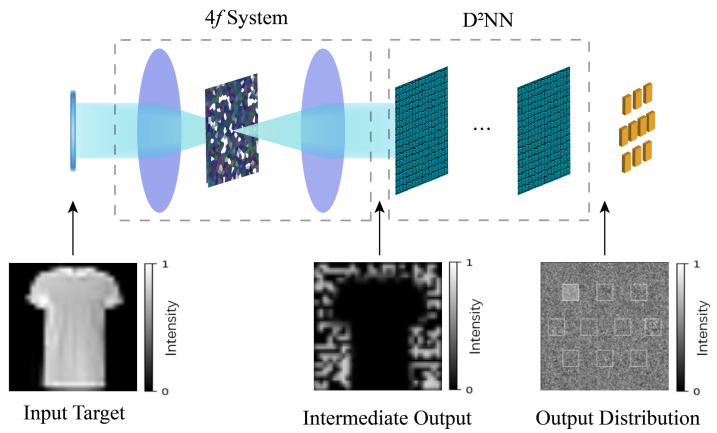
Structure of ODCNN.

**Figure 2 sensors-23-05749-f002:**
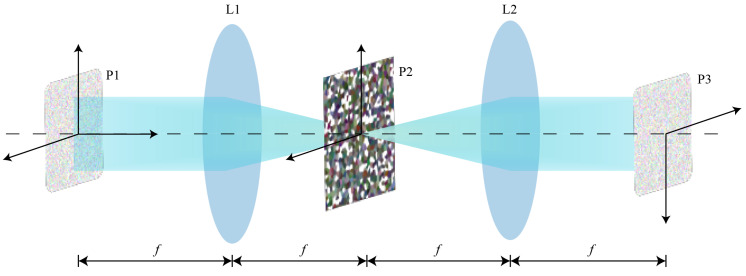
Structure of the optical 4*f* system.

**Figure 3 sensors-23-05749-f003:**
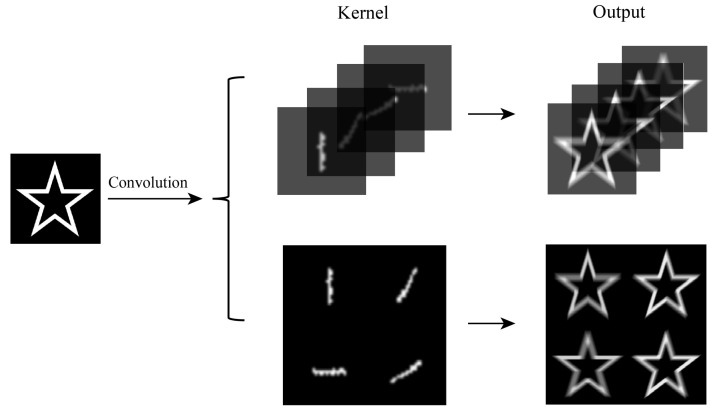
Comparison of convolution kernel arrangement.

**Figure 4 sensors-23-05749-f004:**
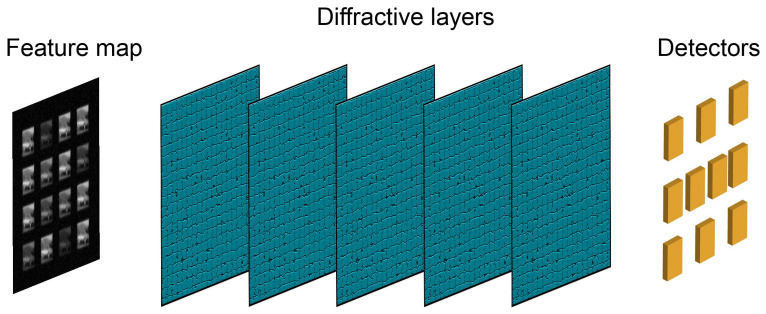
Structure of the fully connected layer.

**Figure 5 sensors-23-05749-f005:**
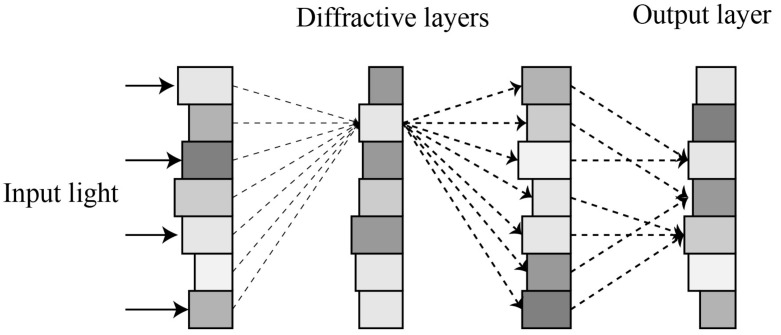
Schematic diagram of network transmission.

**Figure 6 sensors-23-05749-f006:**
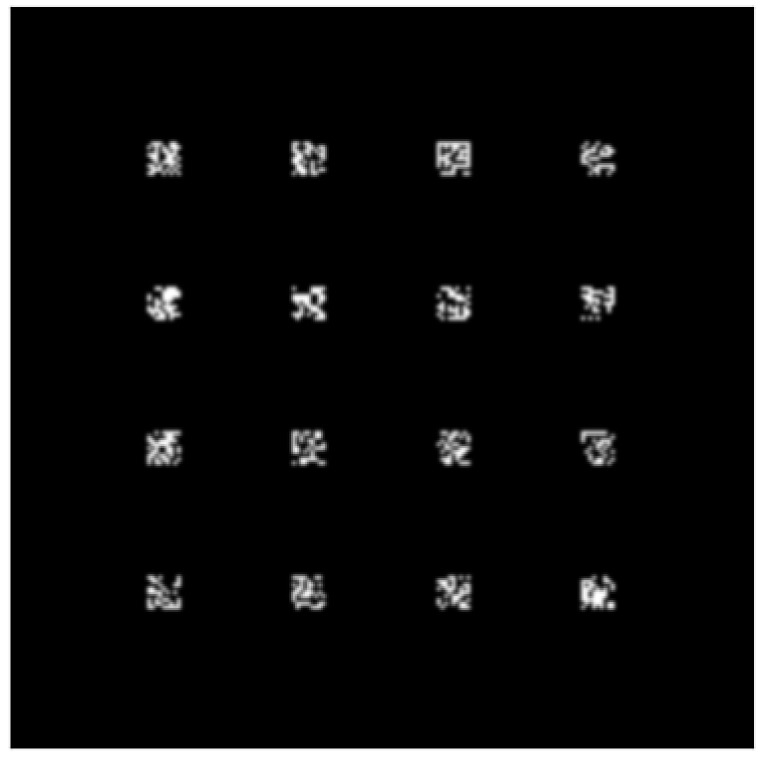
Schematic diagram of convolution kernel distribution.

**Figure 7 sensors-23-05749-f007:**
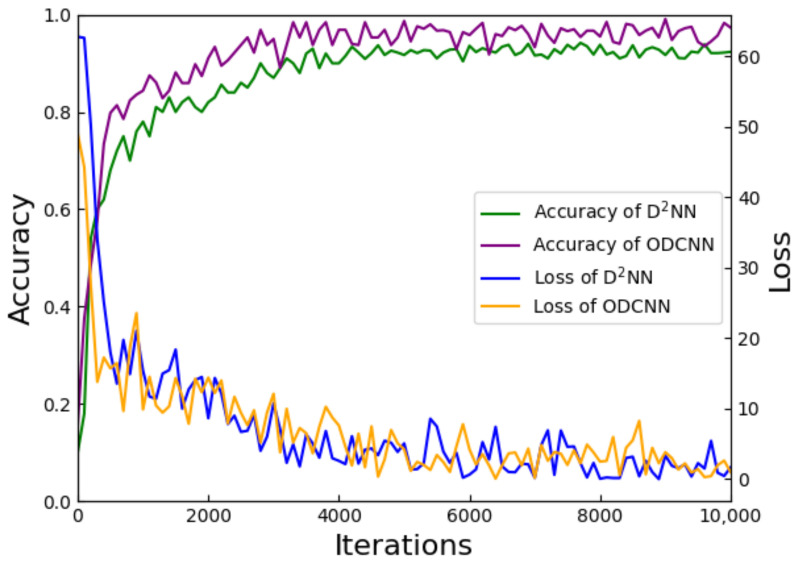
Comparison of classification results and loss curves of ODCNN and D${}^{2}$NN on the MNIST dataset.

**Figure 8 sensors-23-05749-f008:**
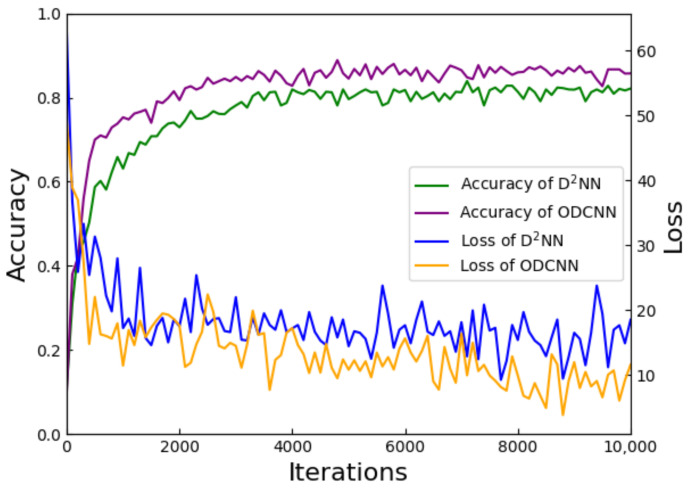
Comparison of classification results and loss curves of ODCNN and D${}^{2}$NN on Fashion–MNIST dataset.

**Figure 9 sensors-23-05749-f009:**
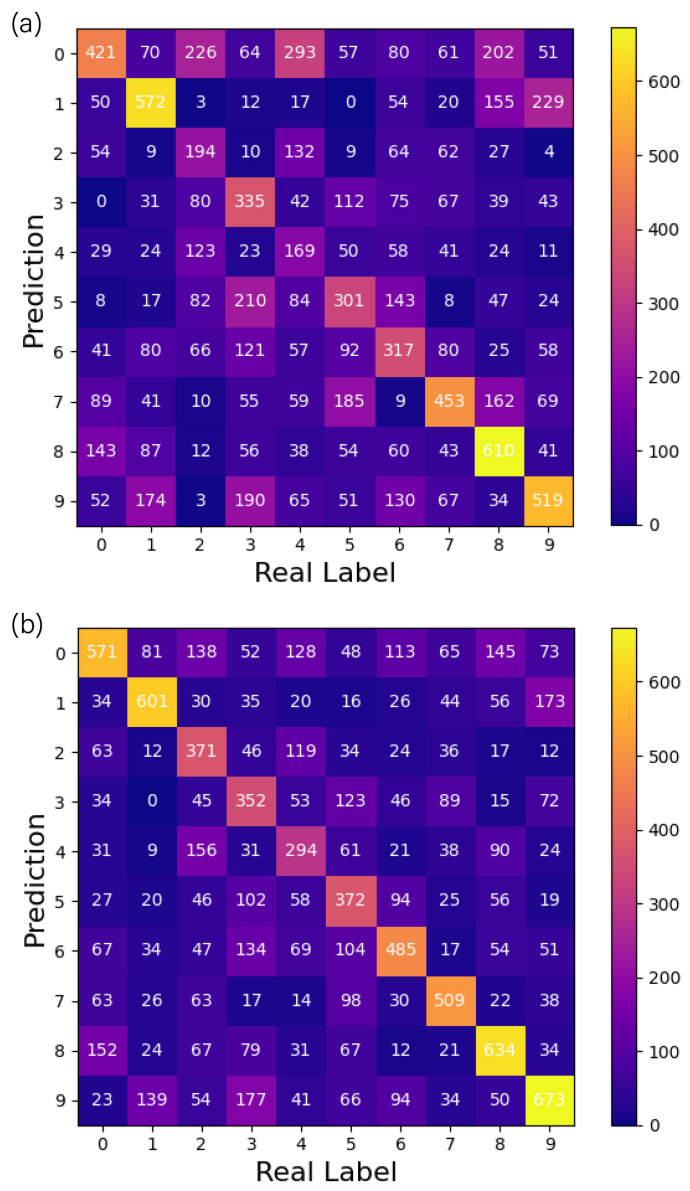
Confusion matrix of (**a**) D${}^{2}$NN and (**b**) ODCNN on the CIFAR-10 dataset.

**Figure 10 sensors-23-05749-f010:**
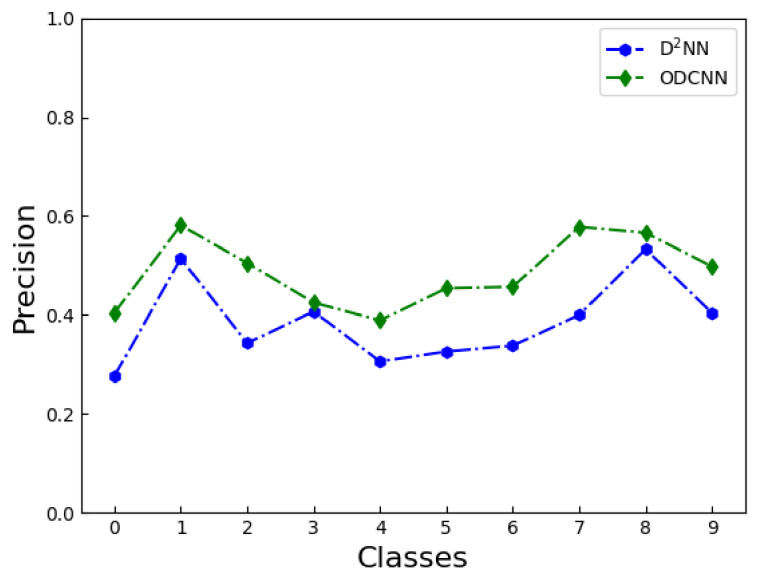
Comparison of the precision of ODCNN and D${}^{2}$NN on the CIFAR-10 dataset.

**Figure 11 sensors-23-05749-f011:**
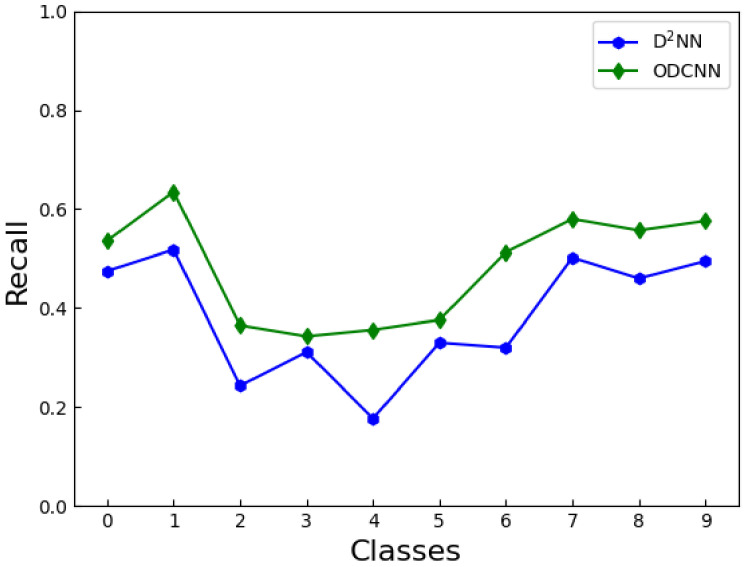
Comparison of the recall of ODCNN and D${}^{2}$NN on the CIFAR-10 dataset.

**Figure 12 sensors-23-05749-f012:**
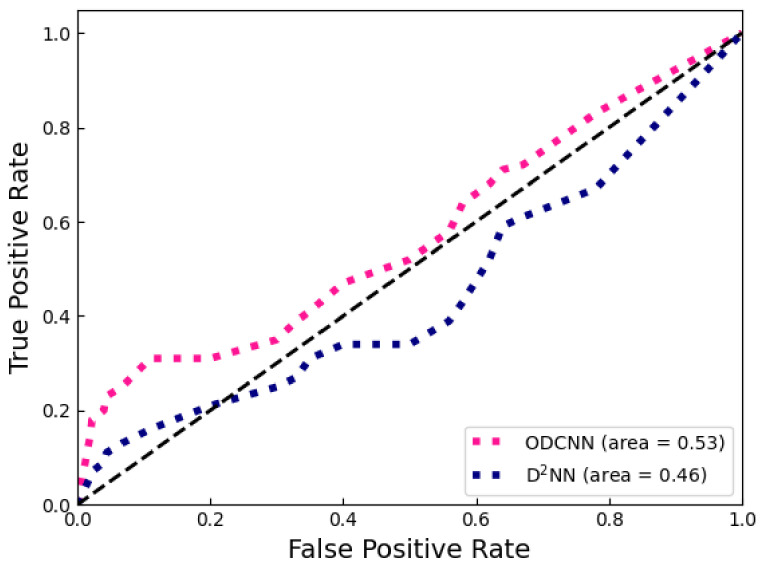
The ROC curves of ODCNN and D${}^{2}$NN on the CIFAR-10 dataset.

**Figure 13 sensors-23-05749-f013:**
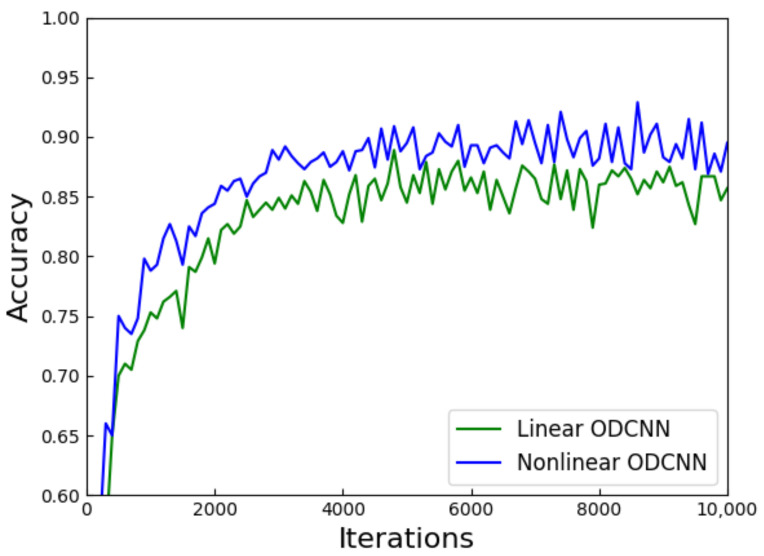
Effect of the nonlinear unit on ODCNN.

**Table 1 sensors-23-05749-t001:** Model structure comparison.

Models		Networks	
D${}^{2}$NN	Input images (200 × 200)	None	5-Optical FC layers (200 × 200)
ODCNN		Opt conv (200 × 200)	5-Optical FC layers (200 × 200)

**Table 2 sensors-23-05749-t002:** Classification accuracy of different models on MNIST and Fashion–MNIST datasets.

Models	Accuracy/%	
	MNIST	Fashion–MNIST
DPU [31]	96.50	84.45
Res-D${}^{2}$NN [17]	98.30	88.42
DNN-NOM [16]	97.04	88.21
OPCNN [29]	96.52	85.20
ODCNN (ReLU)	97.31	87.28
ENN	98.67	89.88

## Data Availability

The data presented in this study are available on request from the corresponding author.

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
