# Peer review of "Optical Diffractive Convolutional Neural Networks Implemented in an All-Optical Way"

_sensors, 2023, doi:10.3390/s23125749_

Round 1
Reviewer 1 Report
The authors propose an optical diffraction convolutional neural network. In the article, the application of the 4f system and diffraction deep neural network is examined.
- The data sets used can be classified with different architectures with high performance. The classification performance of the method is very low. The reason for choosing the model should be detailed.
- There is a class imbalance in the data sets used. Therefore, other performance parameters obtained from the confusion matrix should also be examined.
- The architecture of the model (such as layer parameters) should be detailed. The properties of the layers used should be presented in a table.
-The proposed model and the contributions to the literature should be detailed.
Author Response
Reviewer1
The authors propose an optical diffraction convolutional neural network. In the article, the application of the 4f system and diffraction deep neural network is examined.
Comment 1: The data sets used can be classified with different architectures with high performance. The classification performance of the method is very low. The reason for choosing the model should be detailed.
Response: We gratefully appreciate for your valuable suggestion. In terms of algorithm and model innovation, The MNIST dataset provide us with a unified evaluation standard that can be used to compare the performance of different algorithms on the same task. The simplicity and ease of use of MNIST also facilitates researchers to experiment with new machine learning methods and model architectures, facilitating verification of the feasibility and effectiveness of new methods and providing a foundation for complex vision tasks. The Fashion-MNIST dataset can be seen as an improved version of the MNIST dataset. The expressions related to the selection of the dataset we have updated in the article. Our main objective is the validation of model validity. Optical neural networks are currently in their infancy and there is a gap between their performance and that of traditional network models, which is the direction of our future efforts.
Comment 2: There is a class imbalance in the data sets used. Therefore, other performance parameters obtained from the confusion matrix should also be examined.
Response: Thank you for your suggestions. We added the analysis of Recall and ROC curve (AUC) in our subsequent experiments, and the results proved that the model proposed in this paper has higher classification robustness in multiple categories. The updated content is located at line 271.
Comment 3: The architecture of the model (such as layer parameters) should be detailed. The properties of the layers used should be presented in a table.
Response: We have completely rewritten the relevant expressions. We have included a structural comparison of the models in the article, as shown in Table 1. We describe the series of operations and changes performed by the images into the network according to the contents of the table. Relative to the base D2NN network, we introduce an optical convolutional structure. So the description focuses on the spatial arrangement of the 16 9×9 convolutional kernels in it.
Comment 4:The proposed model and the contributions to the literature should be detailed.
Response: We have enriched the presentation of related contents. For the proposed model, we provide a detailed description of the introduced 4f system and present the data form and the parameter profile of each layer in the network. The specific modifications are in Section 2.1 and in the first paragraph of Chapter 3.

Reviewer 2 Report
1、The present research work in the introduction section was discussed unclearly and the contextual logic is not strong. The content of the introduction must be extensively edited.
2、Insufficient explanation of ODCNN in the paper.
3、what is the 4f system?
4、The experimental part of the paper needs to be supplemented with network performance from similar papers and make comparisons and analysis with them.
Extensive editing of your English language
Author Response
Reviewer2
Comment 1: The present research work in the introduction section was discussed unclearly and the contextual logic is not strong. The content of the introduction must be extensively edited.
Response: Thank you for your comment. We have rewritten the introduction section. First we point out the development bottleneck of electronic computing and introduce optical neural network as an alternative. Then the development status of D2NN and 4f systems are introduced respectively. Finally we illustrate the ODCNN network proposed in this paper.
Comment 2: Insufficient explanation of ODCNN in the paper.
Response: We describe in more detail from the optical convolutional layer and network structure. We introduce the specific composition of the 4f system and the specific parameters of the network structure. The ODCNN can be understood as consisting of one convolutional layer and five fully connected layers. Relevant updates can be found in Section 2.1 and the Experimental Section.
Comment 3: what is the 4f system?
Response: The 4f system is an optical device composed of two convex lenses with the same focal length. The input image is converted into the Fourier domain at the first convex lens,. And it is dot-multiplied with the preset kernel plane. Then the inverse Fourier transform occurs to obtain the output image, thereby completing the convolution operation. The detailed description can be found on line 97.
Comment 4: The experimental part of the paper needs to be supplemented with network performance from similar papers and make comparisons and analysis with them.
Response: We have included a comparative experiment part in the paper. At the end of the experimental part, we compare with optical neural networks such as DPU, Res-D2NN, etc. Experimental results show that ODCNN achieves comparable accuracy to the current best-performing optical network with a concise network structure.

Reviewer 3 Report
The current paper proposes an architecture specific for an Optical Diffractive Convolutional Neural Network implemented in a fully optical manner. The paper is interesting and well written, demonstrating an increased technical and scientific level. However, the following remarks should be taken into account:
(1.) In Introduction, the state of the art should be presented in a more extended manner, by describing more already existing approaches regarding both traditional CNNs and optical neural networks. The original contributions of the authors with respect to the state of the art should be more clearly highlighted.
(2.) More classification performance assessment metrics, such as the recall and the Area under ROC (AuC), should be taken into account.
(3.) More comparisons of the obtained results with other similar results obtained in the state of the art, on the same datasets, are due. For this purpose, the authors should take into account approaches which are based on both traditional and optical neural networks.
The quality of the English language is overall satisfying.
Author Response
Reviewer3
The current paper proposes an architecture specific for an Optical Diffractive Convolutional Neural Network implemented in a fully optical manner. The paper is interesting and well written, demonstrating an increased technical and scientific level. However, the following remarks should be taken into account:
Comment 1: In Introduction, the state of the art should be presented in a more extended manner, by describing more already existing approaches regarding both traditional CNNs and optical neural networks. The original contributions of the authors with respect to the state of the art should be more clearly highlighted.
Response: We appreciate your valuable comments on this paper. We have rewritten the introduction part. Firstly, we highlight the shortcomings of the electronic computation on which traditional neural networks rely. Then the advantages of the optical network are drawn out. The D2NN network and the corresponding improvement work are introduced. And we introduce the application status of the 4f system in the neural network. The contribution of our work mainly lies in the combination of the 4f system and D2NN, which realizes the convolutional neural network in an all-optical way.
Comment 2: More classification performance assessment metrics, such as the recall and the Area under ROC (AuC), should be taken into account.
Response: Thank you for your suggestion. The relevant experimental content is described in line 271. We added recall and ROC curves as new evaluation metrics. Experimental results show that the proposed model has higher recall and more robust classification performance.
Comment 3: More comparisons of the obtained results with other similar results obtained in the state of the art, on the same datasets, are due. For this purpose, the authors should take into account approaches which are based on both traditional and optical neural networks.
Response: In the last part of the experiment, we compared multiple optical neural networks on the MNIST dataset and the Fashion-MNIST dataset. ODCNN has achieved relatively good performance. In addition, we also added an electronic neural network with the same structure as ODCNN. The performance of optical networks is still slightly behind, but its excellent energy consumption and computing efficiency can make up for this gap.
Round 2
Reviewer 1 Report
I recommend publishing the article.
Reviewer 2 Report
The article has been revised according to my revision comments and meets the requirements for publication. I accept the current form.